# Study protocol and design for the assessment of paediatric pneumonia from X-ray images using deep learning

Mark GF Sun,[1] Senjuti Saha [ORCID],[1,2] Syed Ahmar Shah [ORCID],[3] Saturnino Luz [ORCID],[4] Harish Nair,[5] Samir Saha[1]

MGS and SS contributed equally.

[1]Child Health Research Foundation, Dhaka, Bangladesh
[2]International Health, Johns Hopkins University Bloomberg School of Public Health, Baltimore, Maryland, USA
[3]Usher Institute of Population Health Sciences and Informatics, The University of Edinburgh, Edinburgh, UK
[4]Usher Institute of Population Health Sciences and Informatics, School of Molecular Genetic and Population Health Sciences, University of Edinburgh, Edinburgh, UK
[5]School of Molecular Genetic and Population Health Sciences, University of Edinburgh, Edinburgh, UK

**Correspondence to**
Samir Saha; Samir@chrfbd.org

## ABSTRACT

**Introduction** In low-income and middle-income countries, pneumonia remains the leading cause of illness and death in children<5 years. The recommended tool for diagnosing paediatric pneumonia is the interpretation of chest X-ray images, which is difficult to standardise and requires trained clinicians/radiologists. Current automated computational tools have primarily focused on assessing adult pneumonia and were trained on images evaluated by a single specialist. We aim to provide a computational tool using a deep-learning approach to diagnose paediatric pneumonia using X-ray images assessed by multiple specialists trained by the WHO expert X-ray image reading panel.

**Methods and analysis** Approximately 10000 paediatric chest X-ray images are currently being collected from an ongoing WHO-supported surveillance study in Bangladesh. Each image will be read by two trained clinicians/radiologists for the presence or absence of primary endpoint pneumonia (PEP) in each lung, as defined by the WHO. Images whose PEP labels are discordant in either lung will be reviewed by a third specialist and the final assignment will be made using a majority vote. Convolutional neural networks will be used for lung segmentation to align and scale the images to a reference, and for interpretation of the images for the presence of PEP. The model will be evaluated against an independently collected and labelled set of images from the WHO. The study outcome will be an automated method for the interpretation of chest radiographs for diagnosing paediatric pneumonia.

**Ethics and dissemination** All study protocols were approved by the Ethical Review Committees of the Bangladesh Institute of Child Health, Bangladesh. The study sponsor deemed it unnecessary to attain ethical approval from the Academic and Clinical Central Office for Research and Development of University of Edinburgh, UK. The study uses existing X-ray images from an ongoing WHO-coordinated surveillance. All findings will be published in an open-access journal. All X-ray labels and statistical code will be made openly available. The model and images will be made available on request.

## Strengths and limitations of this study

► The main strength of the study is that it will employ a large, high confidence dataset of ~10000 paediatric chest X-ray images interpreted by multiple trained readers for the presence of primary endpoint pneumonia.

► Another strength is that the computational tool will be evaluated against two external paediatric chest X-ray image datasets to assess generalisability of the model.

► The primary limitation of the study is that the computational model will be trained from chest X-ray images derived from a single study site.

countries. Globally, about 138 million cases and 0.9 million deaths are estimated to occur annually due to pneumonia with about six million cases in Bangladesh alone.[1] However, determining the disease prevalence is complicated due to the lack of reliable diagnostic tools: (1) blood cultures are insensitive as most cases are likely not bacteremic and therefore a blood culture cannot determine a bacterial respiratory infection and (2) other diagnostic procedures like radiological findings are difficult to standardise. While the true proportions of pneumonia aetiologies (viral/bacterial) are unknown,[2] 34% of children in Bangladesh diagnosed with pneumonia receive empirical antibiotics.[3] Image-based diagnostics such as the interpretation of chest X-ray images to identify paediatric pneumonia are regarded as the 'gold standard',[4] but they are difficult to standardise and the interpretations can vary between radiologists and clinicians.[5 6] Furthermore, the limited number of well-trained radiologists and clinicians limits the use of X-ray images as a ubiquitous diagnostic tool for pneumonia. This lack of accurate diagnosis hampers the determination of the true burden of pneumonia, and the proportion that is preventable by vaccination, which in turn impedes

## INTRODUCTION
### Paediatric pneumonia overview
Pneumonia is a leading cause of childhood illness and death in developing

institution of evidence-based intervention strategies and vaccine impact measurements.

Deep learning methods such as convolutional neural networks (CNNs) may be used to expedite the adoption of chest X-ray images for the diagnosis of pneumonia. CNNs are composed of multiple feature extractors stacked on top of each other. During the training process, the feature extractors near the model input learn primitive image features such as edges, whereas feature extractors deeper into the model learn domain-specific relationships important for the diagnostic task. CNNs have successfully been applied to assist in the detection and classification of human diseases like breast cancer,[7 8] tuberculosis,[9] diabetic retinopathy[10] and pneumonia[11–13] directly from images. In some instances, these systems performed as well as trained health professionals. While systems have been developed to identify paediatric pneumonia from X-ray images, the datasets used for training are either small (<1000), evaluated on X-ray images derived from the same study site, or do not compare the model's performance with respect to a clinician/radiologist. Our goal is to construct a computational framework to automatically and systematically interpret paediatric chest X-ray images to diagnose pneumonia using ~10 000 training images and whose performance both generalised across datasets and is benchmarked to that of a clinician/radiologist. If successful, an automated system to detect paediatric pneumonia will have significant public health benefits. It will accelerate the adoption of chest X-ray images in low/middle-income country (LMIC) settings, where there is a lack of trained radiologists and clinicians, and contribute to the standardisation of X-ray image analysis. This will encourage clinicians to rationalise the use of antibiotic therapies and to facilitate antibiotic stewardship.

### Diagnosing paediatric pneumonia

Chest X-ray images interpreted to have primary endpoint pneumonia (PEP) are considered the 'gold standard' for the diagnosis of paediatric pneumonia.[4] The PEP case definition was developed by the WHO as part of clinical trials to assess the efficacy of vaccines against *Streptococcus pneumoniae* and *Haemophilus influenza* type b (Hib) and is defined as the presence of end-point consolidation or pleural effusion.[4] Despite efforts to standardise the interpretation of chest radiographs by training specialists to the WHO definitions, significant discrepancies are still observed between readers interpreting the same chest X-ray images, due to the subjective nature of the diagnosis.[5 6] To overcome differences between trained readers, a panel of multiple readers may be employed to interpret images.[6]

Establishing well-funded and rigorous clinical trials with multiple trained readers to evaluate every chest X-ray image is an onerous task. This time intensive task may be alleviated with the aid of computational tools to diagnose pneumonia directly from chest X-ray images. CNN models are one such tool, but require thousands of labelled images to successfully be trained from a randomly

initialized state. To this end, adult radiology reports have been analysed by natural language processing (NLP) systems to programmatically assign labels. This enabled the collection of a large computationally labelled chest X-ray images (40 000–110 000 images) for the training of CNN models to diagnose pneumonia directly from adult chest X-ray images.[11 12 14] Transfer learning can be applied to reduce model training time or to enable model training should the dataset be too small to train a model from a randomly initialised state. Models are first pretrained on a large image dataset such as ImageNet,[15] containing more than a million labelled images of common entities. These pretrained models are transferred to the domain of interest by finetuning the model.[16] Zech *et al* found that transfer learning for a large adult chest X-ray image dataset achieved a performance of 0.931 for the area under the receiver operator curve (auROC). When evaluated on an external benchmark dataset, whose data source differs from that of the training dataset, the model performed significantly worse with an auROC of 0.815. This performance degradation of models trained on a single site was found in three out of five instances,[12] highlighting the need of external evaluation datasets to avoid the reporting of inflated performance metrics, which likely holds true in the paediatric pneumonia setting.

Model training from randomly initialise weights[17 18] and model finetuning has been applied for the diagnosis of paediatric pneumonia directly from X-ray images.[19–21] Efforts to increase performance within the paediatric pneumonia setting have involved using multiple different architectures to form model ensembles, thereby expanding the space of functions that can be learnt to associate an X-ray image to a diagnosis label.[22 23] Chouhan *et al* used transfer learning with five different models[22] based on the AlexNet,[24] DenseNet-121,[25] Inception V3,[26] GoogLeNet[27] and ResNet18[28] architectures, for a paediatric pneumonia X-ray image dataset containing 5856 labelled images.[19] Combining the individual model predictions by a majority vote (99.34 auROC)[22] or in a linear combination scheme (99.76 auROC)[23] were suggestive that a performance increase relative to a single model is possible. Rather than combining the final scalar prediction scores, multiple CNN models may be treated as feature extractors that may be used as input to an attention mechanism, enabling the model to attend to import features. Cha *et al* used the Self-Attention mechanism[29] over the ensemble feature vectors to achieve an auROC of 96.03.[30] Feature selection may be applied over the ensemble feature vector using the minimum redundancy maximum relevance method. The selected features may be used as input into a classifier such as a support vector machine (SVM).[31] Other feature selection schemes include the use of Artificial Ecosystem-based Optimization[32 33] and the Marine Predators Algorithm enhanced by fractional-order calculus,[34] which have respectively been applied for the diagnosis of tuberculosis and COVID-19 from X-ray images. Alternatively, the prediction of a CNN may be gated by an anomaly detection scheme using a

modified loss function to guide model training.[35] These methods were evaluated on images from the same study site; thus, additional investigation is required to ensure that the models are learning discriminatory radiological features of pneumonia and are not exploiting confounding features of the study site.

During CNN model training, image features responsible for the radiological diagnosis of pneumonia are learnt. Rather than learning the image features directly from the training dataset, images features may be explicitly specified. Ke *et al* used global image descriptors of hue, saturation and brightness as input to a fully connected neural network to predict pneumonia.[36] Representative pixels of disease lung tissue were identified using the Moth-Flame[37] and Ant Lion[38] heuristics. Global image descriptors are too coarse of a feature to predict pixel level properties. Mahomed *et al* used Gaussian derivative filters to specify texture features on a per pixel level and used an SVM for classification. A small paediatric dataset of 858 chest X-ray images which was interpreted for PEP by three readers[2] was used to train there model, which achieved a 0.850 auROC.[13] Our study will use 10 000 images interpreted for PEP. Furthermore, by specifying the features that are to be used by a computational prediction method, one may potentially omit features that can be directly learnt from chest X-ray images, which will likely negatively impact model performance.

In light of the grading discrepancies found between readers interpreting paediatric chest X-ray images for pneumonia, our study will use multiple readers trained by the expert WHO X-ray image reading panel. Additionally, all readers will be blinded to the labels provided by the other readers. Furthermore, our study will use X-ray images collected from an LMIC hospital by radiologists for patient services. These images, collected over a period of 8 years, were not part of any specific radiological study. Hence, technologists taking the images were not uniformly trained specifically for this study, reflecting the variability commonly found in most hospitals. To demonstrate that our proposed system generalises to other study sites, we will benchmark the system against images collected from two different sources: (1) the Kumudini Women's Medical College and Hospital (KWMCH) and (2) the WHO Chest Radiography in Epidemiological Studies (WHO CRES) working group.[39]

## Objectives

We aim to build a computational tool using a deep-learning approach to diagnose paediatric pneumonia using X-ray images assessed by multiple WHO-trained specialists. The primary project objectives are to:

1. Collect and label ~10 000 paediatric chest X-rays images with high confidence, indicating if the right and left lung are healthy or if either lung meets the PEP criteria.
2. Develop a software tool leveraging advances in the field of image recognition, in particular deep learning architectures and strategies for training these models, to diagnose paediatric pneumonia from chest X-ray images.
3. Evaluate the model against ~100 images collected from an alternative site labelled by the same readers of this study and against 130 independently collected images from sites differing from those described in this study.[39 40]

## METHODS AND ANALYSIS

The overall design of the study is illustrated in figure 1.

## Participants and data source

Chest X-ray images will be derived from the ongoing WHO-coordinated Invasive Bacterial Vaccine Preventable Diseases (IB-VPD) Surveillance Platform in Bangladesh.[41 42] This surveillance system monitors burden of

Image interpretation Computational modelling

**Figure 1** Study design schema. CRES, Chest Radiography in Epidemiological Studies; KWMCH, Kumudini Women's Medical College Hospital; PEP, primary end point pneumonia.

bacterial pneumonia, meningitis and sepsis in children under 5-year-old inpatients, using protocols described elsewhere.[43] Chest X-ray images are performed on the advice of the treating physicians Digital Imaging and Communications in Medicine images are collected by our team periodically. Technicians performing the chest X-rays are not trained for this specific project avoiding any bias in picture quality. Collection of the chest X-ray images started in 2013 and we expect to reach 10 000 chest X-ray images in 2021.

Two machines are used to capture the images: (1) SG Healthcare Jumong General and 2) TRIUP International Corp TRF100. The intent is for 100 images derived from the KWMCH, also an IB-VPD Surveillance Platform partner, to be included as a test set to assess the impact of images collected from different X-ray machines and technicians. The KWMCH site has two machines to capture images: (1) Hitachi Radnext 50 and (2) General Electric Dx-525. In the IB-VPD surveillance, written consent is obtained from parents or caregivers of all participants for other aspects of the study, including collection of data and additional analysis. Images of any participants that withdraw the consent from the surveillance study will not be considered for this study. The data generated from the individual patients will be deidentified prior to analysis.

### Inclusion criteria

An X-ray image will be included in this study if
► The age of the child for whom the X-ray is performed is <60 months.
► The X-ray was performed in one of the two study hospitals and the child was enrolled in the IB-VPD surveillance study.
► The X-ray image captures the lung area.

The inclusion criteria for the IB-VPD surveillance system for pneumonia are:
► The age of the child is <60 months.
► The child had coughing or difficulty breathing and tachypnoea when calm at a rate of ≥60 breaths/min in an infant aged <2 months, or
► ≥50 breaths/min in an infant aged two to <12 months, or ≥40 breaths/min in children≥12 months.

### Exclusion criteria

Any image that is marked 'uninterpretable' (features of the images are not interpretable with respect to presence or absence of PEP) by the first two readers.

### Sample size

In this study, we are attempting to train computational models rather than to assess if two populations differ. For this reason, we turn to literature to estimate the training dataset sizes required to build the proposed software tool to interpret paediatric chest X-rays. The tool to diagnose paediatric pneumonia (objective 2) will be composed of two stages: (1) lung detection to align the chest X-ray images to a reference image and (2) interpretation of paediatric chest X-rays for the presence of pneumonia.

To identify the location of lungs within the X-ray image, Dai *et al* developed a generative adversarial network and a baseline residual block network. They report Dice coefficient values of 97.3% and 96.3%, respectively, where the Dice coefficient is a measure of similarity between the predicted location and a reference mask, with just 247 adult chest X-rays images.[44] The results will likely translate to the proposed study as we are interested in the identification of the same gross anatomical structures. This suggests that detecting the lung's location within paediatric chest X-rays can readily be achieved with training using only a small proportion (about 300 from 10 000) of the collected chest X-ray images.

For interpretation of chest X rays, we will focus on studies that used expert panels to label X-ray images as a means to estimate the dataset size required to train a CNN to diagnose paediatric pneumonia directly from chest X-ray images. Prior studies used NLP systems to analyse radiology reports to programmatically assign labels, enabling large collections of chest X-ray images to be labelled.[11 12] However, due to the variability found between trained readers when diagnosing paediatric pneumonia from chest X-ray images,[5 6] we believe that a more appropriate strategy is to have each image assessed by multiple readers to ensure a high-quality training dataset. Recent deep learning methods have diagnosed diseases such as tuberculosis after training on 1007 chest X-ray images with a performance of 0.99 for the auROC[9] and diabetic retinopathy with a performance of 0.974 auROC using 128 175 retinal training images (28.1% were classified as referable diabetic retinopathy).[10] For the diagnosis of referable diabetic retinopathy, Gulshan *et al* found that using 10 000 retinal images to train their model resulted in a sensitivity and specificity, respectively, of 97% and 75% with saturation occurring at 60 000 images with a sensitivity and specificity respectively of 97.5% and 93.4%. We noted that the performance found between a reading panel and images labelled by an expert reader with a sensitivity and specificity, respectively, of 77% and 96.3% for paediatric pneumonia.[6]

To date, 8002 paediatric chest X-ray images have been reviewed by a single clinician, of which 2320 (29%) were identified with PEP in either lung, 4786 (60%) had no pneumonia and 898 (11%) were uninterpretable. We expect this class ratio to remain true for the complete dataset. Together, these studies and observations suggest that the 10 000 paediatric chest X-rays being collected will result in a suitable dataset to train a neural network to identify pneumonia cases among paediatric chest X-ray images.

### Study procedure

To build a software tool capable of diagnosing paediatric pneumonia directly from X-ray images, it is first necessary to establish a representative training dataset (objective 1) that is of a sufficiently large size for CNN to be successfully trained (figure 1). To this end, ~10 000 paediatric chest X-rays will be collected and labelled by radiologists/

clinicians standardised to the WHO definitions (each blinded to all labels provided by the other readers)[4] who will interpret the images as having (1) PEP, (2) other lung infiltrates and/or (3) pleural fluid in either the left or right lung, for a total of six possible binary outcomes. Multiple labels for both lungs rather than a single binary value will enable information from both lungs to be leveraged and information about other non-PEP classes may be used to guide the pneumonia prediction task. A challenge to assigning a pneumonia label to an image is the high discordance observed between diagnoses from trained radiologists/clinicians.[5] We plan on mitigating this by having multiple readers standardised to the WHO definitions to read each chest X-ray image. Specifically, two primary readers will assess all images meeting the described inclusion criteria. Chest X-ray images having discordant PEP labels will be reassessed by a third reader. A simple majority voting will be used to assign the multiclass labels for images. The remaining discordant labels (ie, not PEP) will be assigned randomly to reflect uncertainty in the label. Since 'uninterpretable' images may still have evidence of some of the six outcomes, images labelled as 'uninterpretable' by both readers will be excluded from the computational modelling process. Images labelled as 'uninterpretable' by one primary reader but 'interpretable' by the other will be re-evaluated by the third reader (figure 1).

The software tool to diagnose paediatric pneumonia (objective 2) will be composed of two stages: (1) lung detection to align the chest X-ray images to a reference image and (2) interpretation of paediatric chest X-ray images for the presence of pneumonia. Chest X-ray images of paediatric patients can be highly variable, specifically in resource poor settings without well-trained personnel and/or high patient flow, ranging from images of uncentered whole child bodies to properly aligned lung images (figure 2). To aid the paediatric pneumonia prediction task, we first aim to identify the location of the lung. While techniques such as data augmentation can minimise the impact of translation, scaling, and rotation on a neural network's performance, explicitly removing these sources of variation can improve performance.[45] We will evaluate two approaches for their ability to identify the lungs location within an X-ray image. The first model's architecture is a variant of the Visual Geometry Group architecture, but with residual network blocks for pixel-level object detection. This has previously been

shown to work well for adult lung segmentation,[44 46] in combination with image histogram equalisation. This architecture of this lung detection model will be determined from a hyperparameter search, which will match the model's complexity to the lung detection problem. The lung detection model will initially be trained on 300 paediatric chest X-ray images and evaluated on a validation set of an additional 100 paediatric chest X-ray images and assessed using the Dice coefficient to evaluate the model's performance. Additional images may be used if the model's performance is deemed insufficient. In line with observations by Dai et al,[44] 300 images should be sufficient for our lung segmentation and centering task. Key point detection on the segmented images by Binary Robust Invariant Scalable Keypoints[47] or Fast Retina Keypoint[48] to align the images to a reference image using the Open Source Computer Vision Library (OpenCV), an image manipulation toolkit.[49] The second approach for lung detection will be to explore the use of spatial transformer networks[45] in conjunction with object recognition deep learning architectures, thereby enabling a single model to detect the lung and diagnose paediatric pneumonia.

The second stage of the software tool will diagnose pneumonia from the aligned paediatric chest X-ray images. This identification task will be performed in a multitask prediction setting, namely that the CNN's goal will be to predict (1) PEP, (2) other lung infiltrates and/or (3) pleural fluid in both the left and right lung for a total of six binary labels. The pneumonia diagnosis model will be based on the DenseNet architecture.[25] Dilated convolutions[50] will be incorporated into the architecture to aggregate features spanning different distances across the lungs. The final model architecture will be determined from an architecture hyperparameter search over the number of convolution layers per dense block, feature-map growth rate, and feature-map compression will be explored. The number of dense blocks will be kept fixed to four. The binary cross entropy loss function will be used for model training, to assess differences between the model's prediction and the ground truth provided by the labels and to guide how the model parameters are to be updated to improve the model predictions. An ensemble of five models will form the final predictor, whose predictions for each of the six output classes will be the unweighted average of the five models' predicted class probabilities.

## Statistical methods and performance evaluation

To evaluate the quality of the labels associated with the ~10 000 chest X-ray images, we will evaluate the readers' concordance with each other and with the compiled multireader labels using Cohen's kappa coefficient, sensitivity and specificity for both the right and left lung. 95% CIs will be assessed empirically by bootstrapping.

To select the model hyper-parameters (model complexity), the data will be partitioned into five folds and tested on 10% of the data. This will allow us to train

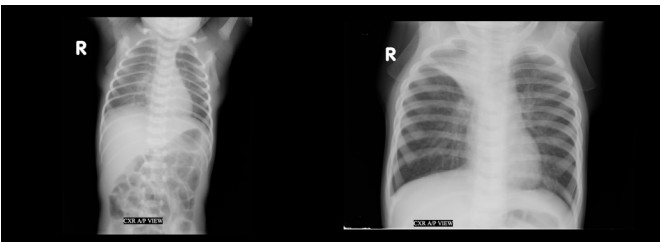

**Figure 2** Digitised chest X-ray images include (left) full-body images and (right) images of just the lung area.

models on 72% (four folds) and validate on 18% (one fold) of the data and to compare models with different hyperparameters. Given the prevalence of pneumonia within this dataset measured to date (29%), having enough positive samples to validate the model for both the right and left lung is a concern, which we believe a five fold cross validation addresses. Hyperparameters will be determined with a mixture of grid searching and prior literature values. These five models will form an ensemble and the final predictions for each of the six output classes will be formed by taking the unweighted average of the predicted class probabilities for each label.

Two external image datasets (objective 3) and 10% of the data derived from primary 10 000 image dataset will be used to evaluate the software tool. The first external test set will be from KWMCH, whose labels will be provided by the sa me readers who labelled the training image dataset. This dataset will enable the study to evaluate the impact of using different X-ray imagining machines and technicians and disease prevalence. The second external test dataset will be from WHO CRES, whose images were independently collected and independently labelled by clinicians and radiologists following the WHO Radiological Pneumonia definitions.[4] This will assess how transferable our method will be to other study sites. Performance will be evaluated using the auROC, the area under the precision-recall curve, sensitivity and specificity. A confusion matrix will also be provided. Finally, 95% CIs will be assessed empirically by bootstrapping. Benchmark dataset will be resampled with replacement 1000 times.

## Patient and public involvement
No patient or members of the public were involved in this study.

## DISCUSSION
The study will be the first to use a large dataset whose labels are derived from multiple expert readers to train a deep-learning model to diagnose paediatric PEP. Prior work to identify pneumonia in adult lungs using a CNN achieved a performance of 0.931 auROC and a 0.815 auROC on an independently collected benchmark. This computational model was trained on chest X-ray images labelled from a single reader whose pneumonia labels were extracted from radiology reports via a NLP system that may introduce extraction errors.[12] A prior study that leverage multiple expert readers, was small in size, containing 858 paediatric chest X-ray images interpreted for PEP by three readers.[2] Since the X-rays were collected over a short period of time, this dataset likely lacks the image variance observed in LMIC settings. The study authors used an SVM to make pixel level predictions and a 95th percentile pixel score as an image score to achieve a 0.850 auROC.[13] We are expecting our model to have similar or better performance since the training data is larger in size and evaluated by multiple readers. Furthermore, the radiological pneumonia definition is broad in

comparison to the more precisely defined WHO of diagnosing PEP.

A limitation of the study is the use of chest X-ray images from individuals <60 months of age derived from a single institution using two machines to capture the images to construct a computational model to interpret chest X-ray images for the presence of pneumonia. Findings from the study may not generalise to individuals ≥60 months of age. However, this is the most appropriate age group for this study as more than 90% of paediatric pneumonia cases are limited to children <5 years of age.

While the computational model will be evaluated against an independently collected and labelled chest X-ray image dataset for the presence or absence of PEP (WHO CRES dataset), some findings may not generalise to other settings (as the training data is from a single institution). To overcome this limitation, we are aiming to extend this study as a multicountry surveillance, if successful.

**Acknowledgements** This research was commissioned by the UK National Institute for Health Research (NIHR) Global Health Research Unit on Respiratory Health (RESPIRE), using UK Aid from the UK Government. The views expressed in this publication are those of the author(s) and not necessarily those of the NIHR or the UK Department of Health and Social Care. We are grateful to Mohammad Shahidul Islam and Md. Shariful Islam for the intellectual and technical guidance.

**Contributors** Conceptual construction of research and design: MGFS, SS and SKS. Critical manuscript review: all authors. All authors have approved the final version of the manuscript for submission for publication.

**Funding** This work has received funding support from NIHR Global Health Research Unit on Respiratory Health (RESPIRE) 16/136/109.

**Competing interests** None declared.

**Patient and public involvement** Patients and/or the public were not involved in the design, or conduct, or reporting, or dissemination plans of this research.

**Patient consent for publication** Not required.

**Provenance and peer review** Not commissioned; externally peer reviewed.

**ORCID iDs**
Senjuti Saha http://orcid.org/0000-0001-6087-6766
Syed Ahmar Shah http://orcid.org/0000-0001-5672-0443
Saturnino Luz http://orcid.org/0000-0001-8430-7875

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
