## [Reviewer comments · BMJ Open]

ARTICLE DETAILS

TITLE (PROVISIONAL)	A study protocol and design for the assessment of paediatric pneumonia from X-Ray images using deep learning
AUTHORS	Sun, Mark; Saha, Senjuti; Shah, Syed Ahmar; Luz, Saturnino; Nair, Harish; Saha, Samir

VERSION 1 – REVIEW

REVIEWER	Robertas Damasevicius Kaunas University of Technology, Lithuania
REVIEW RETURNED	02-Nov-2020

GENERAL COMMENTS	Discussion related deep learning methods for pneumonia recognition should be extended. Especially, focus on the most recent state-of-the-art works, for example, DOI: 10.1016/j.eswa.2019.01.060, DOI: 10.3390/app10020559. Discuss the limitations of these and other works. Research question should be formulated. The study should include more details on the deep learning models to be used. Area under the receiver operator curve is usually abbreviated as AUC rather than AUROC. Using a single metric for evaluation is not enough. Include others such as f1-score. “the data will be partitioned into 5-folds.” – a standard practice is to use 10-fold cross-validation. The planned performance evaluation should also include the use of statistical analysis and statistical tests to calculate the statistical reliability of the results (confidence intervals, p-value).
--

REVIEWER	Eric Scheier Kaplan Medical Center, Israel
REVIEW RETURNED	08-Nov-2020

GENERAL COMMENTS	Reviewer comments: Overall this is an excellent paper that clearly required a great effort to complete, and the final product will help children across the globe. My first comments include corrections of language. This paper could use an English-language editor. There are also a number of phrases that are presumably from the field of computer science (such as “model training regime” “pipeline” “binary cross entropy loss function” “hyper-parameters”). I also think that further clarification is needed on the distinction between infiltrate and bacterial pneumonia (see my multiple comments below). I would be interested in learning more about the disagreements between clinicians who reviewed the x-rays: how many were thought to be thymus, diaphragmatic eventration or hernia, and atelectasis, and
---

	how you expect your model to prospectively handle that distinction. There should also be discussion on the approach to retrocardiac infiltrate – will the system find these on a single view chest x-ray as it finds peripherally-located infiltrates? How will the model adjust for over or under exposed film? There should also be more discussion of the public health implications, as many areas of the world rely on practitioners who are not physicians for pediatric care, and the transferability of your work to point of care ultrasound, which is also gaining traction as an alternative to chest x-ray in both the developing and in the developed world. Given the size of this project, a separate section in ‘methods’ titled ‘statistics’ should be added and should describe the statistical software and methods used. There should be a more thorough discussion of current literature including Reference #12, which appears to be very similar to your project. Line 74: I am not clear what bacterial cultures have to do with the diagnosis of community associated pneumonia (CAP). CAP is primarily a clinical, not radiologic, diagnosis. I recommend that authors look at the following citations:  1. Harris M, Clark J, Coote N, et al. British Thoracic Society guidelines for the management of community-acquired pneumonia in children: update 2011. Thorax. 2011;66 Suppl 2:ii1-ii23. 2. Bradley JS, Byington CL, Shah SS, et al. The management of community-acquired pneumonia in infants and children older than 3 months of age: clinical practice guidelines by the Pediatric Infectious Diseases Society and the Infectious Diseases Society of America. Clin Infect Dis. 2011;53(7):e25-e76. 3. Florin TA, Gerber JS. Sticking by an Imperfect Standard: Chest Radiography for Pediatric Community-Acquired Pneumonia. Pediatrics. 2020;145(3):e20193900. doi:10.1542/peds.2019-3900 Line 74: delete “relatively” and add a citation showing me that this is true. Line 75: “children in Bangladesh diagnosed with pneumonia. Line 76: “Frontal” is not to my knowledge an accepted term for the PA/AP chest x-ray. Line 78: Interpretation of CXR is always subjective. I would rewrite that “interpretation can vary between radiologists and clinicians.” Line 80: radiologists/clinicians should be radiologists and clinicians. “restricts” should be “limits”. Line 81: “The diagnosis of pneumonia.” Line 88: Delete “such.” Line 89: Delete “evidence-based case management.” One of the ideas that needs clarification in this paper is that your system identifies pulmonary infiltrates. The diagnosis of pneumonia is made by the clinician using clinical context together with (or without) the chest x-ray. Line 94: Your “gold standard” statement requires a citation. Line 114: should be clarified to read that the performance of the computational model was 0.931 AUROC. Lines 163,168: why limit the study to children less than 60 months of age? Presumably because the IB-VPD study limited to this age group, but this should be clarified here as it was in line 309. Alternately, describe the limitations of the IB-VPD study earlier in the text. A discussion of IB-VPD is warranted because it is a screening mechanism to avoid missing potentially dangerous bacterial pneumonia but will capture every viral infiltrate in every child with RSV bronchiolitis. Which is why I feel that it is important to clarify earlier that your study builds a model that will identify
--	--

	infiltrates in the lung that may or may not be consistent with bacterial pneumonia. Line 181: is “rotation align” a computer science term? Line 224: the first “lung” appears to be a typo Line 230: “between the two readers in either the right or left lung, when considering the lungs independently” can be deleted. Line 243: “images of the only whole lung”. Did you mean a single hemithorax? Line 256: “Key point detection on the segmented images by BRISK [27] or FREAK [28] to align the images to a reference image using OpenCV.” It would be helpful to have an index of these computer science acronyms, and to reference the company that makes them as you would hardware. Line 281: “WHO Radiological Pneumonia definitions” needs a citation that includes these definitions, preferably earlier in the text. Line 286: IRB approval was noted earlier in the methods section and does not need to be repeated here Line 290: what is github? Line 292: how was the public involved? Lines 319-322: should be in a ‘limitations’ section in the discussion rather than in the conclusion. Limitations would seem to include that the accuracy of the model depends on the quality of the hardware used to acquire the images.
--	--

VERSION 1 – AUTHOR RESPONSE

Reviewer: 1

Discussion related deep learning methods for pneumonia recognition should be extended. Especially, focus on the most recent state-of-the-art works, for example, DOI: 10.1016/j.eswa.2019.01.060, DOI: 10.3390/app10020559. Discuss the limitations of these and other works.

Response: We have expanded the discussion of pneumonia prediction methods to include the aforementioned recent works and other associated works in the “Diagnosing paediatric pneumonia” section. We have attempted to highlight the importance of 1) evaluating computational models on a benchmark dataset that is independently collected from the one used for training, 2) a dataset that captures the image variations observed within low- and middle income countries (LMIC) settings, and 3) the importance of directly learning features from the X-ray image datasets rather than making explicit assumptions about (lines 113– 173).

Research question should be formulated.

Response: Thank you for this comment. We have edited our “Objective” section to clarify this (lines 175 – 187).

The study should include more details on the deep learning models to be used.

Response: We have provided additional details on the deep learning models, with special focus on the diagnostic model in the “Study Procedure” section.

Area under the receiver operator curve is usually abbreviated as AUC rather than AUROC.

Response: While AUC is the dominant abbreviation, since we will be using the area under the precision-recall curve (auPR) as an additional measure, we are of the opinion that auROC would be beneficial to the reader as it will clarify which area under the curve we will be referencing when discussing the model’s performance.

Using a single metric for evaluation is not enough. Include others such as f1-score.

Response: We appreciate this comment. Evaluations will be performed using the area under the receiver operator curve (auROC), area under the precision-recall curve (auPR), sensitivity, and specificity (lines 360 - 362).

“the data will be partitioned into 5-folds.” – a standard practice is to use 10-fold cross-validation.

Response: We agree that 10-fold cross-validation is the norm for most problem domains using machine learning methods that have only a few parameters that need to be tuned. However, with deep learning models often containing millions of parameters, 10-fold cross-validation is computationally expensive. Furthermore, our X-ray image dataset has a pneumonia prevalence of 29%, which includes both the right and left lung. Having enough positive pneumonia samples to monitor training progression is important to ensure performance stability. Taken together, we believe 5-fold cross-validation is appropriate for this domain problem and for the computational model that is to be trained (line 344 – 352).

The planned performance evaluation should also include the use of statistical analysis and statistical tests to calculate the statistical reliability of the results (confidence intervals, p-value).

Response: Thank you for this comment. For all considered evaluation methods, 95% confidence intervals will be empirically assessed by bootstrapping (line 362 - 363).

Reviewer: 2

Reviewer comments:

Overall this is an excellent paper that clearly required a great effort to complete, and the final product will help children across the globe.

My first comments include corrections of language. This paper could use an English-language editor.

There are also a number of phrases that are presumably from the field of computer science (such as “model training regime” “pipeline” “binary cross entropy loss function” “hyper-parameters”).

I also think that further clarification is needed on the distinction between infiltrate and bacterial pneumonia (see my multiple comments below).

I would be interested in learning more about the disagreements between clinicians who reviewed the x-rays: how many were thought to be thymus, diaphragmatic eventration or hernia, and atelectasis, and how you expect your model to prospectively handle that distinction.

There should also be discussion on the approach to retrocardiac infiltrate – will the system find these on a single view chest x-ray as it finds peripherally-located infiltrates?

How will the model adjust for over or under exposed film?

There should also be more discussion of the public health implications, as many areas of the world rely on practitioners who are not physicians for pediatric care, and the transferability of your work to point of care ultrasound, which is also gaining traction as an alternative to chest x-ray in both the developing and in the developed world.

Given the size of this project, a separate section in ‘methods’ titled ‘statistics’ should be added and should describe the statistical software and methods used.

There should be a more thorough discussion of current literature including Reference #12, which appears to be very similar to your project.

Response: Thank you. We find these constructive criticisms very helpful.

We have tried to edit the paper for further clarity and have avoided jargon where possible. If the term was deemed important for others to understand assumptions made by the model, we paid specific attention to explaining these terms.

We have included a statistical analysis section within the performance evaluation section (lines 339 - 363).

We have included a section on public health implication (lines 96 - 100).

Our tool will not be transferrable for point of care ultrasound, as the model is being trained on X-ray images. However, other studies from other groups are underway to build such models for ultrasound also, which is very exciting.

We have also tried to clarify throughout the manuscript that the objective of our project is to build a computational tool to detect pneumonia. As none of the variables fed into the model included

diagnoses like diaphragmatic eventration or hernia, atelectasis, etc, these will not be picked up by the tool. The output of the model will be the 6 variables (eg. right PEP) that are used to train the model. No additional output, like peripherally-located infiltrates, will be detected.

In addition, X-ray images that are considered uninterpretable by the readers are not included in dataset for this study (lines 227 – 229, 290 – 292).

To overcome the saturation effects of underexposed or overexposed X-ray images, all images will undergo histogram equilibration. This is a common vision strategy to improve image contrast by increasing the dynamic range of pixel intensities observed within an image.

Line 74: I am not clear what bacterial cultures have to do with the diagnosis of community associated pneumonia (CAP). CAP is primarily a clinical, not radiologic, diagnosis. I recommend that authors look at the following citations:

1. Harris M, Clark J, Coote N, et al. British Thoracic Society guidelines for the management of community-acquired pneumonia in children: update 2011. *Thorax*. 2011;66 Suppl 2:ii1-ii23.
2. Bradley JS, Byington CL, Shah SS, et al. The management of community-acquired pneumonia in infants and children older than 3 months of age: clinical practice guidelines by the Pediatric Infectious Diseases Society and the Infectious Diseases Society of America. *Clin Infect Dis*. 2011;53(7):e25-e76.
3. Florin TA, Gerber JS. Sticking by an Imperfect Standard: Chest Radiography for Pediatric Community-Acquired Pneumonia. *Pediatrics*. 2020;145(3):e20193900. doi:10.1542/peds.2019-3900

Response: Thank you for these citations. These were very helpful in revising the manuscript.

Line 74: delete “relatively” and add a citation showing me that this is true.

Response: Revised accordingly.

Line 75: “children in Bangladesh diagnosed with pneumonia.

Response: Revised accordingly.

Line 76: “Frontal” is not to my knowledge an accepted term for the PA/AP chest x-ray.

Response: Revised accordingly.

Line 78: Interpretation of CXR is always subjective. I would rewrite that “interpretation can vary between radiologists and clinicians.”

Response: Revised accordingly.

Line 80: radiologists/clinicians should be radiologists and clinicians. “restricts” should be “limits”.

Response: Revised accordingly.

Line 81: “The diagnosis of pneumonia.”

Response: This sentence has been revised for clarity.

Line 88: Delete “such.”

Response: Revised accordingly.

Line 89: Delete “evidence-based case management.” One of the ideas that needs clarification in this paper is that your system identifies pulmonary infiltrates. The diagnosis of pneumonia is made by the clinician using clinical context together with (or without) the chest x-ray.

Response: Revised accordingly.

Line 94: Your “gold standard” statement requires a citation.

Response: Thank you. We have added a citation. “Cherian T, Mulholland EK, Carlin JB, et al. Standardized interpretation of paediatric chest radiographs for the diagnosis of pneumonia in epidemiological studies. Bull World Health Organ 2005;83:353–359”.

Line 114: should be clarified to read that the performance of the computational model was 0.931 AUROC.

Response: We have more clearly indicated that the model achieved a performance of 0.931 auROC on the test dataset. We have used the term external benchmark dataset to indicate the datasets used to assess a model's generalization performance to study sites.

Lines 163,168: why limit the study to children less than 60 months of age? Presumably because the IB-VPD study limited to this age group, but this should be clarified here as it was in line 309. Alternately, describe the limitations of the IB-VPD study earlier in the text. A discussion of IB-VPD is warranted because it is a screening mechanism to avoid missing potentially dangerous bacterial pneumonia but will capture every viral infiltrate in every child with RSV bronchiolitis. Which is why I feel that it is important to clarify earlier that your study builds a model that will identify infiltrates in the lung that may or may not be consistent with bacterial pneumonia.

Response: Thank you for this comment. We have clarified in the revised manuscript that the WHO IB-VPD specifically enrolls children with signs and symptoms of bacterial pneumonia. We have included the IB-VPD inclusion criteria for pneumonia (lines 235 - 240). The WHO RSV inclusion criteria are broader and not included in this study.

Line 181: is "rotation align" a computer science term?

Response: We have replaced this with "align" to aid readability of the text.

Line 224: the first "lung" appears to be a typo

Response: Thank you for this comment. We have edited the sentence for clarity.

Line 230: "between the two readers in either the right or left lung, when considering the lungs independently" can be deleted.

Response: Revised accordingly.

Line 243: "images of the only whole lung". Did you mean a single hemithorax?

Response: Thank you for this comment. We have edited the sentence for clarity. We mean that some images contain uncentered whole child bodies, where the lungs are a small part of the whole image, while some are proper and aligned images of the lungs.

Line 256: "Key point detection on the segmented images by BRISK [27] or FREAK [28] to align the

images to a reference image using OpenCV.” It would be helpful to have an index of these computer science acronyms, and to reference the company that makes them as you would hardware.

Response: Thanks for the comment. Since these acronyms appear only once in the text, we believe describing them inline would be the most beneficial to the reader. In subsequent manuscripts where these terms appear multiple times throughout the text, we'll consider the use of a reference index for these computer science acronyms.

Line 281: “WHO Radiological Pneumonia definitions” needs a citation that includes these definitions, preferably earlier in the text.

Response: Thank you. We have added a citation.

Line 286: IRB approval was noted earlier in the methods section and does not need to be repeated here

Response: Thank you for this comment. We have removed this section.

Line 290: what is github?

Response: GitHub is a repository used by developers around the world to manage their codes. Its appeal is that anyone can download a copy of the code to reproduce the study results.

Line 292: how was the public involved?

Response: No patient or public were involved in this study.

Lines 319-322: should be in a ‘limitations’ section in the discussion rather than in the conclusion. Limitations would seem to include that the accuracy of the model depends on the quality of the hardware used to acquire the images.

Response: According to the editorial suggestion, we have deleted the conclusion part. The limitation of our tool is now mentioned as part of the discussion.

VERSION 2 – REVIEW

REVIEWER	Robertas Damasevicius
----------	-----------------------

	Silesian University of Technology, Poland
REVIEW RETURNED	13-Jan-2021

GENERAL COMMENTS	This study protocol paper aims to provide a computational tool based on deep learning to diagnose paediatric pneumonia using a large set of X-ray images assessed by multiple specialists. Comments:  1. Introduction section should explain the rationale for the study protocol and what knowledge gap it may fill. Appropriate previous literature should be referenced. Unfortunately, the paper fails to properly analyze and review the deep learning methods used to tackle the research problem. The number of analyzed papers is too small and mostly outdated. The most recent state-of-the-art papers must be discussed in this rapidly evolving research field, including the papers that use more complex architectures such as hybrid deep learning/nature optimized methods and ensemble models. For example, “A novel method for detection of tuberculosis in chest radiographs using artificial ecosystem-based optimisation of deep neural network features”, “COVID-19 image classification using deep features and fractional-order marine predators algorithm” could be discussed, among others. 2. Methods and analysis: the cross-validation procedure discussed in the paper is incorrect. The dataset must be split into three subsets: training, testing and validation. The models are trained on the training dataset and tested on the testing dataset. Finally, the selected model with the best hyperparameter values is evaluated on the validation dataset. 3. Performance evaluation should also include the confusion matrix. 4. L.364: provide more details about how the bootstrapping procedure will be used. 5. The limitations of the methodology and possible threats-to-validity of the results must be addressed.
---

REVIEWER	Eric Scheier Kaplan Medical Center, Israel
REVIEW RETURNED	11-Jan-2021

GENERAL COMMENTS	I don't think that reference #10 is required as it is not from the peer-reviewed medical literature. it should be noted in the text that reference 11 is looking at x-rays from an adult population, reference 12 appears to be the only such study is from a relatively small pediatric population and therefore your research is novel and adds to existing literature. this is made clear in 'strengths and limitations' and again in 'discussion' but i think it's an important point and should be stated explicitly early in the text itself. line 159: not clear why reference [2] is needed. line 169: 'not trained in any specific manner'. i'm sure they were trained, perhaps the training was not uniform. line 382: 'better defined', need to clarify better than what
---

VERSION 2 – AUTHOR RESPONSE

Reviewer: 1

Prof. Robertas Damaševičius, Kaunas University of Technology

Comments to the Author:

This study protocol paper aims to provide a computational tool based on deep learning to diagnose paediatric pneumonia using a large set of X-ray images assessed by multiple specialists.

Comments:

1. Introduction section should explain the rationale for the study protocol and what knowledge gap it may fill.

Response: We've highlighted the studies' strengths relative to previous work in the introduction. (Line 101-105)

Appropriate previous literature should be referenced. Unfortunately, the paper fails to properly analyze and review the deep learning methods used to tackle the research problem. The number of analyzed papers is too small and mostly outdated. The most recent state-of-the-art papers must be discussed in this rapidly evolving research field, including the papers that use more complex architectures such as hybrid deep learning/nature optimized methods and ensemble models. For example, "A novel method for detection of tuberculosis in chest radiographs using artificial ecosystem-based optimisation of deep neural network features", "COVID-19 image classification using deep features and fractional-order marine predators algorithm" could be discussed, among others.

Response: The manuscript further extends the literature background in section "Diagnosing paediatric pneumonia". (Line 118–190)

2. Methods and analysis: the cross-validation procedure discussed in the paper is incorrect. The dataset must be split into three subsets: training, testing and validation. The models are trained on the training dataset and tested on the testing dataset. Finally, the selected model with the best hyperparameter values is evaluated on the validation dataset.

Response: We thank the reviewer for clearly defining the test and validation dataset terminology, as their meaning is often swapped. For this comment we'll use the reviewer's definitions. For the paper we'll use the definition that the validation set is used to select the best hyperparameters and that the test set is used to derive the model's performance. The reason to adhere to the above definition is that the authors believe it most closely follows the wording of "cross-validation", where it is clear that the training and validation set is used to tune the model, whereas the test dataset is used to derive performance measurements.

If only a single dataset from a single source is available, the three-way data split outlined by the reviewer is mandatory in order to assess the model's generalization ability on the validation set. This study will use two validation / benchmark image datasets originating from sites other than the Dhaka Shishu Hospital (DSH) site, from which the primary dataset is derived. Attaining performance metrics on datasets having a pixel distribution differing from the one used for training is more meaningful as it will more closely represent the model's performance if applied to an X-ray image dataset at other sites. As such the utility of a three-way split of the primary dataset from DSH to create a validation set whose pixel distribution will be similar to that of the training set is markedly diminished. Performance

measurements from the two external validation / benchmark datasets will outweigh performance measurements derived from the validation dataset. Another consideration is the primary dataset size, while large, is still small relative to other vision datasets. Further splitting the primary dataset will reduce the amount of data available for training. Consequently, a thoughtful balance is required. In the original manuscript we proposed omitting the validation set. If the reviewer's concern is to assess performance degradation between the validation set and the two-external benchmark / validation datasets (which will occur) then we propose using a small validation set of 10% of the primary source data. (Line 362, 371 – 372)

3. Performance evaluation should also include the confusion matrix.

Response: Thank you. The confusion matrix will be incorporated as part of the evaluation measurements. (Line 380 - 382)

4. L.364: provide more details about how the bootstrapping procedure will be used.

Response: We've clarified that the bootstrapping procedures will resample the evaluation datasets with replacement 1000 times. (Line 382)

5. The limitations of the methodology and possible threats-to-validity of the results must be addressed.

Response: We have addressed the study limitations (study participants < 60 months from training images from a single site) in "Strengths and limitations of this study" and have also mentioned them in the "Discussion".

Reviewer: 2

Dr. Eric Scheier, Kaplan Medical Center

Comments to the Author:

I don't think that reference #10 is required as it is not from the peer-reviewed medical literature.

Response: Reference #10 is highly cited due it being the first time a large number of images were used to train a deep learning model and compared to radiologists. The manuscript now also includes the retrospective PLoS Medicine companion paper (reference [14]).

it should be noted in the text that reference 11 is looking at x-rays from an adult population,

Response: We've highlighted that the reference uses image from an adult population within section "Diagnosing paediatric pneumonia". (Line 122-126)

reference 12 appears to be the only such study is from a relatively small pediatric population and therefore your research is novel and adds to existing literature. this is made clear in 'strengths and limitations' and again in 'discussion' but i think it's an important point and should be stated explicitly early in the text itself.

Response: The manuscript now emphasizes the study's size in comparison to other studies using the PEP definition throughout the manuscript. (Line 93-97, 171-174)

line 159: not clear why reference [2] is needed.

Response: The reference was included since the study evaluates what the dominant causes of pneumonia are (viral vs bacterial). This is the largest study on causes of pediatric pneumonia ever conducted. We did insert the reference next to the comment "blood culture cannot determine a bacterial respiratory infection", which we have removed in this revision.

line 169: 'not trained in any specific manner'. i'm sure they were trained, perhaps the training was not uniform.

Response: The intention was the lack of training uniformity among technologists. We've incorporated the reviewer's wording. (Line 184-186)

line 382: 'better defined', need to clarify better than what

Response: Thank you. In the revised manuscript, the WHO primary-end point pneumonia definition is more precisely defined. (Line 400-401)

VERSION 3 – REVIEW

REVIEWER	Robertas Damaševičius Silesian University of Technology, Poland
REVIEW RETURNED	22-Feb-2021
GENERAL COMMENTS	Well revised.